# The Metabolic Effects of *Cynara* Supplementation in Overweight and Obese Class I Subjects with Newly Detected Impaired Fasting Glycemia: A Double-Blind, Placebo-Controlled, Randomized Clinical Trial

**DOI:** 10.3390/nu12113298

**Published:** 2020-10-28

**Authors:** Mariangela Rondanelli, Antonella Riva, Giovanna Petrangolini, Pietro Allegrini, Luisa Bernardinelli, Teresa Fazia, Gabriella Peroni, Clara Gasparri, Mara Nichetti, Milena Anna Faliva, Maurizio Naso, Simone Perna

**Affiliations:** 1IRCCS Mondino Foundation, 27100 Pavia, Italy; mariangela.rondanelli@unipv.it; 2Unit of Human and Clinical Nutrition, Department of Public Health, Experimental and Forensic Medicine, University of Pavia, 27100 Pavia, Italy; 3Research and Development Unit, Indena, 20139 Milan, Italy; antonella.riva@indena.com (A.R.); giovanna.petrangolini@indena.com (G.P.); pietro.allegrini@indena.com (P.A.); 4Department of Brain and Behavioral Science, University of Pavia, 27100 Pavia, Italy; luisa.bernardinelli@unipv.it (L.B.); teresa.fazia01@ateneopv.it (T.F.); 5Endocrinology and Nutrition Unit, Azienda di Servizi alla Persona “Istituto Santa Margherita”, University of Pavia, 27100 Pavia, Italy; clara.gasparri01@universitadipavia.it (C.G.); dietista.mara.nichetti@gmail.com (M.N.); milena.faliva@gmail.com (M.A.F.); mau.na.mn@gmail.com (M.N.); 6Department of Biology, Sakhir Campus, College of Science, University of Bahrain, Sakheer P.O. Box 32038, Bahrain; simoneperna@hotmail.it

**Keywords:** *Cynara*, impaired fasting glucose, insulin sensitivity, obesity, overweight

## Abstract

Impaired fasting glucose (IFG) is a condition that precedes diabetes and increases the risk of developing it. Studies support the hypoglycemic effect of *Cynara*
*scolymus* (Cs) extracts due to the content of chlorogenic acid, which is a potent inhibitor of glucose 6-phosphate translocase and of dicaffeoylquinic acid derivatives that modulate the activity of alpha-glucosidase. Given this background, we investigated whether a new highly standardized Cs extract could improve glycemic control, insulin sensitivity and other metabolic parameters (total cholesterol (TC), low-density lipoprotein-cholesterol (LDL-C), high-density lipoprotein-cholesterol (HDL-C) Triglycerides, Apolipo protein B (ApoB), Apolipo protein A (ApoA), waist circumference, visceral adipose tissue (VAT) by dual-energy X-ray absorptiometry (DXA) in overweight subjects with newly diagnosed IFG. Fifty-four subjects (females/males 26/28, mean ± SD age 51.5 ± 6.2) were randomly assigned to the supplemented group (*n* = 27) and placebo (*n* = 27). After multiple testing correction, statistically significant interactions between time and group were observed for the primary endpoint glycemia (β = 0.36, *p* < 0.0001) and for the secondary endpoints HDL (β = −0.10, *p* < 0.0001), total cholesterol/HDL (β = 0.27, *p* < 0.0001), LDL (β = 0.15, *p* = 0.005), LDL/HDL (β = 0.23, *p* = 0.001), insulin (β = 1.28, *p* = 0.04), glycated hemoglobin (β = 0.21, *p* = 0.0002), A1c-derived average glucose (β = 0.34, *p* = 0.0002), ApoB (β = 6.00, *p* = 0.01), ApoA (β = −4.50, *p* = 0.04), ApoB/ApoA (β = 0.08, *p* = 0.003), waist circumference (β = 1.89, *p* = 0.05), VATβ = 222.37, *p* = 0.005). In conclusion, these results confirm that Cs supplementation has a significant effect on metabolic parameters in IFG patients.

## 1. Introduction

The risk of developing type 2 diabetes increases if the metabolic condition defined as impaired fasting glycemia (IFG) is present [1].

Most subjects with IFG will develop type 2 diabetes within 10 years: specifically, an increase in type 2 diabetes in 70% of men and 40% of women with IFG over a 10-year period has been demonstrated, as compared to normo-glycemic subjects [2].

Various natural compounds derived from plant extracts, spices, herbs and essential oils have demonstrable benefit in the management of subjects with IFG [3].

Frequently, hypoglycemic herbal extracts have been used as an alternative remedy to glucose-lowering drugs, which can have side effects such as low blood sugar levels, lactic acidosis, idiosyncratic liver cell injury, permanent neurological deficit, digestive discomfort, headache and dizziness [4]. Hypoglycemic herbal extracts are widely used as non-prescription treatment for diabetes [3].

Among hypoglycemic extracts, *Cynara scolymus* (Cs) extract has received a lot of attention; there are now in vitro studies and animal models that confirm this activity. The in vitro study by Arion and the animal studies by Matsui and by Fantini support the hypoglycemic effect of *Cynara* extracts [5,6,7]. This activity seems to be mainly related to the content of chlorogenic acid in Cs [5,6,7,8]. This compound is a potent inhibitor of glucose 6-phosphate translocase, an essential component of the hepatic glucose 6-phosphatase system which regulates the homeostasis of blood glucose [5]. In addition, dicaffeoylquinic acid derivatives of Cs can also play an hypoglycemic role in modulating the activity of alpha-glucosidase and consequently the catabolism of dietary carbohydrates [6].

The favorable effects of Cs extracts on serum glucose regulation have also been clearly demonstrated in humans. A study showed the efficacy of artichoke extract (600 mg/day) on the reduction of glycometabolic parameters in overweight subjects with IFG. Specifically, the supplemented group with *Cynara* had significant decreases in fasting blood glucose (−9.6%), homeostatic metabolic assessment (−11.7%), glycosylated hemoglobin (−2.3%), A1c-derived average glucose (AGAD) (−3.1%) and lipidic pattern, while the placebo group did not show any significant difference. Compared with the placebo, considering inter-group analysis, the supplemented group showed a significant difference in fasting blood glucose, homeostatic metabolic assessment and lipidic pattern [9]. However, it is important to underline that this study was conducted with a different extract of *Cynara*, obtained from Cs flowering heads instead of leaves and characterized by a high content of caffeoylquinic acid [9].

In the study of Kuczmannova et al., *Cynara cardunculus* demonstrated good anti-glucosidase, anti-glycation and anti-hyperglycemic effects [10]. Nutraceutical could be a very interesting strategy to counteract metabolic disease. Ostan et al. revealed that nutraceutical supplementation in association with a balanced Mediterranean diet can significantly improve the inflammatory status in elderly people; in particular, the authors demonstrated that d-Limonene supplementation could improve insulin resistance parameters, such as glucose and insulin levels and Homeostasis Model Assessment for Insulin Resistance (HOMA-IR) index [11].

Moreover, the formulation of functional foods represents a promising and interesting scenario. In the study of Colantuono et al., an artichoke stem powder (ASP) was used as a food ingredient in the formulation of new breads [12]. Artichoke stem was selected as a promising functional ingredient for its content of dietary fiber and polyphenols, mainly including caffeoylquinic acids (CQA), di-caffeoylquinic acids (DCQA) and flavones. The gastrointestinal digestion of the ASP-enriched breads was performed in vitro. The polyphenol-rich extracts resulting from the digestion of ASP-enriched breads exerted inhibitory activity towards α-glucosidase that was proportional to the ASP content of the breads [12]. More recently, Mare et al. demonstrated that a brioche enriched with wheat bran and bergamot fiber could be a novel useful tool in preventing diabetes and controlling the glycometabolic status of type 2 diabetic patients [13].

Given this background, we investigated whether a new highly triple standardized Cs extract from leaves that maintain the natural bouquet of the plant with the following specifications (HPLC title)—caffeoylquinic acids ≥ 5.0%; flavonoids ≥ 1.5%; cynaropicrin ≥ 1.0%—could improve glycemic control, insulin sensitivity and other metabolic parameters, such as total cholesterol (TC), HDL cholesterol (HDL-C), LDL cholesterol (LDL-C), triglycerides (TG), Homeostasis Model Assessment (HOMA), apolipoprotein B (ApoB), apolipoprotein A (ApoA), waist circumference and visceral adipose tissue (VAT) by dual-energy X-ray absorptiometry (DXA), in overweight subjects with newly diagnosed IFG.

## 2. Materials and Methods

### 2.1. Population

A randomized, double-blind, placebo-controlled trial was conducted in overweight and obese (Body Mass Index (BMI) 25–35 kg/m^2^) men and women with newly detected IFG (blood glucose between 6.1 and 7.0 mmol/L, glycosylated hemoglobin <7.0%), as defined by the American Diabetes Association [14], consequently admitted, as outpatients, to the Dietetic and Metabolic Unit of the “Santa Margherita” Institute, University of Pavia, Italy.

These adult subjects (aged between 45 and 55 years) were included in this randomized, placebo-controlled clinical trial between September 2018 and January 2020.

The subjects, without history of cardiovascular disease (CVD), were not taking any medication likely to affect glucose or lipid metabolism (oral hypoglycemic agents and statins) and were free of overt liver, renal and thyroid disease. The subjects who smoked or drank more than two standard alcoholic beverages/day (20 g of alcohol/day) were excluded from the study. Physical activity was recorded. Sedentary subjects were admitted to the study. The experimental protocol was approved by the Ethics Committee of the University of Pavia (ethical code number: 9321/14122018) and was registered at ClinicalTrials.gov. All the volunteers gave their written informed consent.

### 2.2. Dietary Supplement

The dietary treatment was associated with 2 daily oral doses (before lunch and dinner) of 500 mg of Cs tablets. The supplementation period was 8 weeks. The tablets (for the requirement of 4 weeks, 120 tablets in total) were delivered at the time of the first blood sample.

Tablets containing 500 mg of artichoke extract (triple standardized to contain caffeoylquinic acids ≥ 5.0%; flavonoids ≥ 1.5%; cynaropicrin ≥ 1.0%, by HPLC) and placebo were provided by Indena SpA (Milan, Italy). Tablets with no active ingredient were used as placebo and were identical to Cs ones in terms of size, shape, color, odor and taste. Cs and placebo film-coated tablets had similar composition in terms of inactive food-grade components. Before release, the film-coated tablets were tested for appearance, average mass, uniformity of mass, HPLC-content of Cs active compounds, disintegration time and microbiological quality. All procedures were performed according to Food Supplement European Regulation.

Adherence to treatment was assessed by counting the number of supplements remaining when the participants returned to the laboratory. A value of 90% of the total tablets consumption of the supplementation was achieved.

Identical products for each treatment group were assigned to a subject number according to a coded (AB) block randomization table prepared by an independent statistician. Investigators were blinded to the randomization table, the code assignments and the procedure. Independently by supplementation (Cs or placebo tablets), the subjects followed a similar low-energy diet. Regarding blinding, the active intervention and placebo were given in identical containers devoid of any labeling by the principal investigator, who was not involved in any of the assessments.

### 2.3. Adverse Events

Adverse events were based on spontaneous reporting by subjects, as well as open-ended inquiries by members of the research staff. Moreover, routine blood biochemistry parameters (creatinine, liver function) were evaluated at the start and at the end of supplementation.

### 2.4. Glycemic and Lipidic Parameters

The glycemic and lipidic parameters were assessed before the start of the study at baseline (t0), after 30 days (t1) and after 60 days at the end of treatment (t2). In order to avoid venipuncture stress, blood samples were obtained through an indwelling catheter inserted in an antecubital vein. Blood samples were immediately centrifuged and stored at −80 °C until assayed. Fasting blood glucose (FBG), total cholesterol (TC), low-density lipoprotein-cholesterol (LDL-C), high-density lipoprotein-cholesterol (HDL-C) and triglyceride (TG) levels were measured by automatic biochemical analyzer (Hitachi 747, Tokyo, Japan). Serum concentration of hemoglobin A1c (HbA1c) was determined by high-performance liquid chromatographic method using automatic HbA1c analyzer (Tosoh HLC-723G7, Japan). A1c-derived average glucose (ADAG) was calculated [15]. The serum insulin was evaluated by a double antibody RIA (Kabi Pharmacia Diagnostics AB, Uppsala, Sweden) and expressed as pmol/L. The intra- and inter-assay coefficients of variation were below 6% and the low detection limit was 10.7 pmol/L. To determine insulin resistance, subjects were instructed to fast for 12 h before obtaining the blood sample. Furthermore, the subjects refrained from any form of physical exercise for 48 h before the blood sampling. Insulin resistance was evaluated using the Homeostasis Model Assessment (HOMA) [16].

### 2.5. Anthropometric Mesaurements and Dietary Counseling

Nutritional status was assessed using anthropometric measurements at start of the study at baseline (t0) and after 60 days at the end of supplementation (t2). Body weight and height were measured following a standardized technique [17] and the BMI was calculated (kg/m^2^). Anthropometric parameters were always collected by the same investigator.

Subjects were trained to restrict their daily energy intake by a moderate amount, 3344 kJ/d less than daily requirements based on WHO criteria (World Health Organization, 1985, Washington DC, USA), with a regimen that maintained a prudent balance of macronutrients: 25–30% of energy from fat (cholesterol < 200 mg), 55–60% of energy from carbohydrates (10% from simple carbohydrates), with 25 g of bran and 15–20% of energy from protein. A registered dietician performed initial dietary counseling. A 3-day weighed-food record of 2 weekdays and 1 weekend day was performed during the first and the last week of the study. Dietary records were analyzed using a food-nutrient database (Rational Diet, Milan, Italy).

### 2.6. Body Composition

Body composition (fat free mass (FFM), fat mass (FM) and gynoid and android fat distribution) was measured by dual-energy X-ray absorptiometry (DXA) with the use of a Lunar Prodigy DXA (GE Medical Systems). The in vivo coefficients of variation (CVs) were 0.89% and 0.48% for whole body fat (FM) and FFM, respectively.

Visceral adipose tissue volume was estimated using a constant correction factor (0.94 g/cm^3^). The software automatically places a quadrilateral box, which represents the android region, outlined by the iliac crest and with a superior height equivalent to 20% of the distance from the top of the iliac crest to the base of the skull [18].

### 2.7. Primary and Secondary Endpoints

We considered as primary endpoint glycemia and as secondary endpoints total cholesterol, HDL, total cholesterol/HLD, LDL, LDL/HDL, triglycerides, insulin, glycated hemoglobin, A1c-derived average glucose, HOMA, ApoA, ApoB, ApoB/ApoA, systolic blood pressure, diastolic blood pressure, liver function tests (aspartate transaminase (AST), alanine transaminase (ALT), gamma-glutamyl transferase (GGT)), creatinine, waist circumference, VAT, fat mass and lean mass.

Except for systolic blood pressure, diastolic blood pressure, waist circumference, VAT, lean and fat mass, which were measured at two time points (t_0_ and t_1_), all the other endpoints were collected at three different time points, i.e., t_0_, t_1_ and t_2_.

### 2.8. Statistical Analysis

Sample size was calculated on the basis of the results from Rondanelli et al. [9] in which a glucose reduction of −0.61 mmol/L with a 95% confidence interval (−0.89; −0.33) for 8 weeks of intervention was observed. Considering two balanced groups with 1:1 allocation (n1 = n2), an alpha significance level set at 0.01 and 90% of power in detecting differences between the groups, for an effect size d = 0.61, a sample size of 44 subjects in total (22 subjects per arm) was estimated. Differences between groups at baseline were investigated in each continuous variable using *t*-tests for independent data. To evaluate statistically significant changes over time for primary and secondary endpoints within and between the two groups, we fitted linear mixed models with time, group and the interaction time*group as fixed effect, including a random effect in the form of 1|subject in order to take into account intra-subject correlation produced by the repeated measurements [19]. Normality of residuals was assessed graphically and with Shapiro–Wilk test. All the models were adjusted for age, sex and BMI. Benjamini–Hochberg correction, fixing the false discovery rate (FDR) at α < 0.05, was used to account for multiple comparison [20].

Descriptive statistics are reported as mean ± standard deviation (SD). All analysis was performed on R 3.5.1 statistical software using the nlme and stats packages [21,22].

## 3. Results

A total of 54 subjects, 26 females and 28 males, with mean (±SD) age of 51.5 ± 6.27 were randomly assigned to the supplemented group (*n* = 27, 14 females and 12 males) and placebo (*n* = 27, 12 females and 15 males). Figure 1 shows the flow chart of the study. There were no dropouts.

Baseline characteristics of participants are shown in Table 1. No statistically significant differences were observed between the two groups in baseline characteristics.

Table 2 reports the descriptive statistics for each endpoint measured in the two groups at baseline (t_0_), after 30 days (t_1_) and after 60 days (t_2_).

Linear mixed models, adjusted for sex, age and BMI, were fitted to evaluate significant pre- and post-treatment changes (time) between the two groups on each continuous primary and secondary endpoint (full results are reported in Table 3).

After multiple testing correction, statistically significant interactions between time and group (meaning that the change in score over time is different for each group) were observed for the primary endpoint glycemia (β = 0.36, *p* < 0.0001) and for the secondary endpoints HDL (β = −0.10, *p* < 0.0001), total cholesterol/HDL (β = 0.27, *p* = 0.00002), LDL (β = 0.15, *p* = 0.005), LDL/HDL (β = 0.23, *p* = 0.001), insulin (β = 1.28, *p* = 0.04), glycated hemoglobin (β = 0.21, *p* = 0.0002), A1c-derived average glucose (β = 0.34, *p* = 0.0002), ApoB (β = 6.00, *p* = 0.01), ApoA (β = −4.50, *p* = 0.04), ApoB/ApoA (β = 0.08, *p* = 0.003), waist circumference (β = 1.89, *p* = 0.05) and VAT (β = 222.37, *p* = 0.005). As for the within-group differences, at the end of the intervention, the supplemented group showed a statistically significant decrease in glycemia (β = −0.39, *p* < 0.0001), total cholesterol (β = −0.03, *p* = 0.003), total cholesterol/HDL (β = −0.23, *p* < 0.0001), LDL (β = −0.09, *p* = 0.03), LDL/HDL (β = −0.19, *p* < 0.0001), insulin (β = −1.76, *p* = 0.001), glycated hemoglobin (β = −0.18, *p* < 0.0001), A1c-derived average glucose (β = −0.28, *p* < 0.0001), HOMA (β = −0.71, *p* < 0.0001), ApoB (β = −3.72, *p* = 0.04), ApoB/ApoA (β = −0.06, *p* = 0.002), waist circumference (β = −3.33, *p* < 0.0001), VAT (β = −210.74, *p* = 0.003) and fat mass (β = −1772.63, *p* = 0.003) and a statistically significant increase in HDL (β = 0.08, *p* < 0.0001) and ApoA (β = 3.74, *p* < 0.0001). No statistically significant changes from baseline were observed in the placebo group. These results are shown in Table 4.

In Figure 2 are graphically summarized the within- and between-group supplementation effects with respect to the glycemia parameters (i.e., fasting glycemia, insulin, glycated hemoglobin, a1c-derived average glucose). There were no substantial side effects among the participants.

## 4. Discussion

The main interesting result of this clinical study regards the primary endpoint: after multiple testing corrections, statistically significant interactions between time and group were observed for glycemia. In addition, the other parameters concerning glucose metabolism (glycated hemoglobin, A1c-derived average glucose and insulin) showed a significant interaction between time and group. These results confirmed a previous randomized study conducted with the extract of Cs flowering heads extract [9], although it is important to note that, in this case, we used a new, highly standardized Cs extract (triple standardized as HPLC title: caffeoylquinic acids ≥ 5.0%; flavonoids ≥ 1.5%; cynaropicrin ≥ 1.0%) that maintains the natural bouquet of Cs.

In recent years, there has been growing evidence that plant-food polyphenols, due to their biological properties, may be used as unique nutraceuticals and supplementary treatments of diabetes, as their beneficial effects were largely attributed to phenolic compounds, including flavonoids, phenolic acids, lignans and stilbenes on metabolic disorders and complications induced by dysregulation of glucose metabolism [23].

These compounds, abundantly present in *Cynara*, gained much attention due to their antioxidant activities, which potentially have beneficial implications for human health; in particular, polyphenol fractions are able to enhance glucose uptake into muscle cells and to regulate key enzymes implicated in hepatic glucose output, which could decrease glycemia levels [24].

Animal investigations showed the potential effects of polyphenols and flavonoids contained in *Cynara* to modulate carbohydrate and lipid metabolisms, decrease glycemia and insulin resistance, increase lipid metabolism and counteract oxidative stress. In particular, a recent animal study demonstrated that the ability of an ethanol extract from leaves of *Cynara* to reduce blood glucose levels could be attributed to different synergistic mechanisms, such as improvement of peripheral sensitivity to remnant insulin and inhibition of intestinal disaccharidase, like α- amylase activity. These properties could affect the glucose absorption rate in the small intestine, explained as the hypoglycemic effects of artichoke leaves [25].

In addition, studies in vitro demonstrated that flavonoids exert a good inhibiting action against α- amylase [26]; possess an inhibitory effect on glucose absorption due to competitive inhibition of sodium-dependent glucose transporter-1 [27]; increase glucose uptake in peripheral tissues (insulin-like activity); and reduce the intestinal absorption of glucose by digestive enzyme inhibition, repairing damaged cells and stimulating the β cells to insulin secretion [28].

Considering that the risk of developing type 2 diabetes increases if IFG is present [1], and that several diabetes prevention trials have shown that strategies that reduce the rate of conversion to diabetes can also modify CVD risk factors [29], the possibility of improving the parameters of carbohydrate metabolism by consuming a Cs dietary supplement represents a great opportunity for the management of subjects with IFG, thus preventing the possible development of diabetes.

Regarding the secondary endpoints of the study on lipid metabolism, after multiple testing correction, statistically significant interactions between time and group were observed for HDL, total cholesterol/HDL, LDL, LDL/HLD, ApoB, ApoA and ApoB/ApoA.

The mechanism of this hypocholesterolemic action may be due to the inhibition of dietary cholesterol absorption in the intestine, its production by the liver or stimulation of the biliary secretion of cholesterol and cholesterol excretion in the feces, as demonstrated in animal models [25].

Other studies in animal models demonstrated the potential for lipid-lowering and anti-atherogenic effects of aqueous extract from leaves of Cs, as shown by the decrease in total cholesterol and LDL and accompanied by a decrease in inflammatory markers of IL-1, TNF-α, IFN, CRP, oxidized LDL and anti-oxidized-LDL [30].

Results of our study on lipid metabolism in individuals with untreated moderate hypercholesterolemia are in agreement with previous studies, in which artichoke extracts were used alone [31,32] or in combination with other botanicals [33,34,35]. Of note is that, in this supplementation study, we used a new highly standardized Cs extract that preserves the whole natural bouquet of the leaves.

In addition, the data concerning the anthropometric evaluation and the body composition show an improvement of the metabolic status: statistically significant interactions between time and group were observed for waist circumference and VAT.

This is the first clinical study published in the literature that evaluates VAT quantitative changes in subjects with IFG following the intake of a dietary supplement.

Visceral fat is highly metabolically active and is constantly releasing free fatty acids (FFA) into the portal circulation; as such, visceral fat content contributes to various features of the metabolic syndrome, such as hyperinsulinemia, systemic inflammation, dyslipidemia and atherosclerosis [36].

The significant reduction in VAT weight could have a pivotal role in the improvement in glycemic and lipid patterns observed in this study. VAT is an independent predictor of endogenous insulin sensitivity and glucose tolerance; this, in turn, suggests that reducing VAT is crucial for improving insulin sensitivity and preventing diabetes in high-risk individuals [37]. 

Finally, no side effect has been observed during the trial.

The main limitation of this study is that statistical analysis does not allow us to demonstrate a cause–effect relationship between the various parameters and, therefore, we can only hypothesize the central role of VAT reduction in improving metabolic and glycemic parameters.

Another limitation of this study is that the TT genotype of TCF7L2-rs7903146 polymorphism has not been evaluated. It has been demonstrated that transcription factor 7-like 2 (TCF7L2)-rs7903146 polymorphism is associated with increased risk of type 2 diabetes and the response of insulin. Insulin resistance to artichoke leaf extract (ALE) may be affected by TCF7L2-rs7903146 polymorphism [38,39].

Regarding the strengths of this study, the first is that the Cs extract used was highly standardized and characterized by a high content of caffeoylquinic acid and flavonoids, expressed as luteolin glycosides. The issue of the standardization, characterization, preparation and toxicity of an herbal extract is crucial [40].

Another strength is the attention paid to the choice of the patients evaluated: overweight and obese (body mass index (BMI) 25–35 kg/m^2^) men and women with newly detected IFG (blood glucose between 6.1 and 7.0 mmol/L, glycosylated hemoglobin < 7.0%), as defined by the American Diabetes Association.

In conclusion, the results of this study indicate that bioactive food ingredients could produce significant beneficial actions in the management of newly detected IFG by reducing metabolic parameters, such as glycemia, total cholesterol/HDL, LDL, LDL/HLD, insulin, glycated hemoglobin, A1c-derived average glucose, ApoB, ApoB/ApoA.

## 5. Conclusions

In the short term, the Cs leaf extract has been shown to be well tolerated and effective. The results confirm that Cs supplementation has a significant pivotal role in improving metabolic parameters in overweight IFG individuals. Further studies with a larger number of subjects and in other categories of IFG individuals (not just in overweight people with newly detected IFG) are clearly needed to confirm the effect of Cs extract on IFG.

## Figures and Tables

**Figure 1 nutrients-12-03298-f001:**
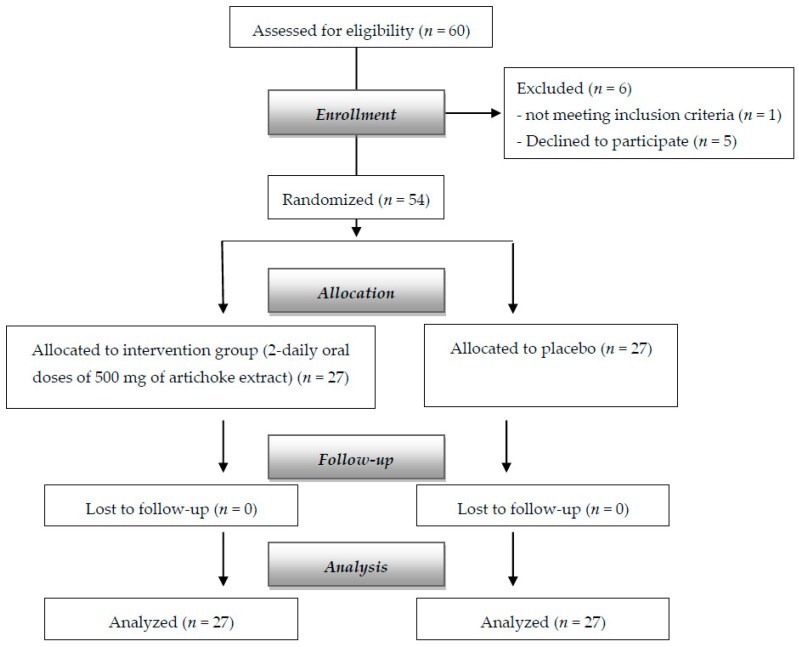
Flow diagram of the study.

**Figure 2 nutrients-12-03298-f002:**
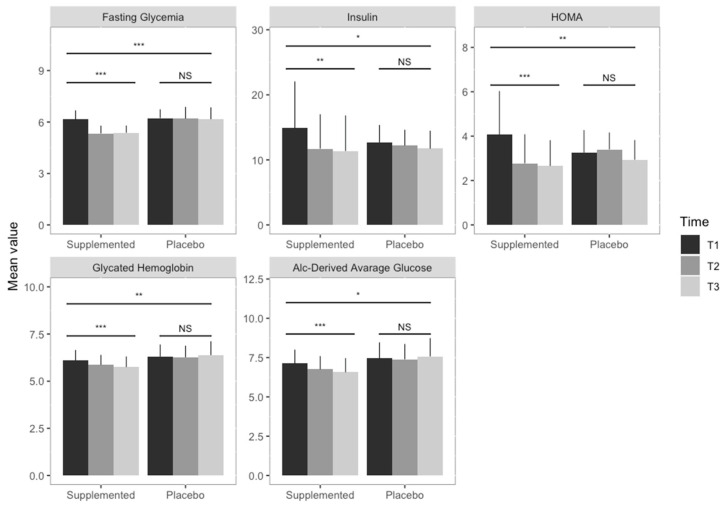
Supplementation effects on glycemia parameters. Bar graphs illustrate the between- and within-group supplementation effect on glycemia parameters. Each graph shows the mean values of the glycemia parameters in the two groups (i.e., supplemented and placebo) in the three different time points (grey scales), with error bars representing the standard deviation. Triple asterisks indicate a significant difference (*p* < 0.0001), double asterisks a significant difference (*p* < 0.001), while the single asterisk represents a statistically significant difference (*p* < 0.05). Results obtained fitting linear mixed effect models indicate a statistically significant decrease in fasting glycemia, insulin, HMA, glycated hemoglobin and A1c-derived average glucose in the supplemented group, while no statistically significant results (NS) were observed in the placebo group. Statistically significant differences were observed between the two groups for the investigated parameters.

**Table 1 nutrients-12-03298-t001:** Baseline (t_0_) age, BMI, primary and secondary endpoints mean ± (SD) and *p*-value for t-test of the difference between the two groups.

	Supplemented (*n* = 27) Mean (SD)	Placebo (*n* = 27) Mean (SD)	*p*-Value
Age (years)	51.44 (6.64)	51.55 (6.01)	0.95
BMI (kg/m^2^)	28.95 (3.59)	29.72 (2.53)	0.36
***Primary endpoint***			
Fasting Glycemia (mmol/L)	6.15 (0.53)	6.23 (0.51)	0.59
***Secondary endpoints***			
Total Cholesterol (mg/dL)	5.43 (0.53)	5.35 (0.56)	0.59
HDL (mg/dL)	1.51 (0.48)	1.49 (0.55)	0.89
Total Cholesterol/HDL	3.90 (1.12)	4.08 (1.52)	0.62
LDL (mg/dL)	3.29 (0.57)	3.23 (0.86)	0.75
LDL/HDL	2.45 (0.99)	2.62 (1.38)	0.61
Triglycerides (mg/dL)	1.35 (0.49)	1.36 (0.36)	0.93
Insulin (mcU/mL)	14.87 (7.19)	12.69 (2.68)	0.15
Glycated Hemoglobin (%)	6.10 (0.56)	6.30 (0.64)	0.22
A1c-Derived Average Glucose (mmol/L)	7.12 (0.89)	7.45 (1.03)	0.22
HOMA	4.08 (1.96)	3.26 (1.01)	0.06
ApoB (mg/dL)	119.78 (18.27)	117.89 (13.77)	0.67
ApoA (mg/dL)	123.04 (16.95)	120.11 (22.63)	0.59
ApoB/ApoA	0.99 (0.21)	1.01 (0.21)	0.73
Systolic Blood Pressure (mmHg)	124.42 (6.68)	123.52 (7.31)	0.64
Diastolic Blood Pressure (mmHg)	81.54 (5.62)	80.37 (6.19)	0.47
AST (UI/L)	18.48 (5.14)	18.37 (3.94)	0.93
ALT (UI/L)	19.48 (5.06)	20.22 (3.84)	0.55
GGT (U/L)	18.19 (7.20)	19.44 (6.42)	0.50
Creatinine (mg/dL)	0.83 (0.07)	0.86 (0.12)	0.41
Waist Circumference (cm)	108.68 (10.79)	106.89 (12.63)	0.58
VAT (g)	1693.37 (871.50)	1849.41 (851.68)	0.51
Fat Mass (g)	32,620.00 (12,068.42)	36,380.78 (10,132.98)	0.22
Lean Mass (g)	43,296.41 (8917.54)	45,429.04 (6867.50)	0.33

Abbreviations—BMI: Body Mass Index, HDL: High-density lipoprotein Cholesterol, LDL: Low-density lipoprotein Cholesterol, HOMA: HOMA index, ApoB: Apolipoprotein B, ApoA: Apolipoprotein A, AST: Aspartate Transaminase, ALT: Alanine Transaminase, GGT: Gamma Glutamyl Transferase, VAT: Visceral Adipose Tissue.

**Table 2 nutrients-12-03298-t002:** Parameter values for each endpoint measured in the two groups at baseline (t_0_), after 30 days (t_1_) and after 60 days (t_2_).

	Supplemented Group (*n* = 27)			Placebo Group (*n* = 27)		
	t_0_	t_1_	t_2_	t_0_	t_1_	t_2_
***Primary endpoint***						
Fasting Glycemia (mmol/L)	6.15 (0.53)	5.33 (0.46)	5.37 (0.43)	6.23 (0.51)	6.20 (0.68)	6.17 (0.69)
***Secondary endpoints***						
Total Cholesterol (mg/dL)	5.43 (0.53)	5.29 (0.68)	5.37 (0.55)	5.35 (0.56)	5.41 (0.63)	5.42 (0.63)
HDL (mg/dL)	1.51 (0.48)	1.61 (0.52)	1.68 (0.50)	1.49 (0.55)	1.45 (0.54)	1.46 (0.53)
Total Cholesterol/HDL	3.90 (1.12)	3.57 (1.04)	3.44 (0.95)	4.08 (1.52)	4.18 (1.38)	4.17 (1.36)
LDL (mg/dL)	3.29 (0.57)	3.11 (0.67)	3.11 (0.54)	3.23 (0.86)	3.34 (0.84)	3.34 (0.87)
LDL/HDL	2.45 (0.99)	2.18 (0.95)	2.07 (0.88)	2.62 (1.38)	2.70 (1.24)	2.69 (1.23)
Triglycerides (mg/dL)	1.35 (0.49)	1.26 (0.43)	1.28 (0.41)	1.36 (0.36)	1.33 (0.34)	1.36 (0.37)
Insulin (mcU/mL)	14.87 (7.19)	11.72 (5.29)	11.36 (5.48)	12.69 (2.68)	12.20 (2.44)	11.73 (2.76)
Glycated Hemoglobin (%)	6.10 (0.56)	5.87 (0.53)	5.75 (0.57)	6.30 (0.64)	6.27 (0.62)	6.38 (0.74)
A1c-Derived Average Glucose (mmol/L)	7.12 (0.89)	6.75 (0.84)	6.56 (0.90)	7.45 (1.03)	7.39 (0.98)	7.57 (1.17)
HOMA	4.08 (1.96)	2.77 (1.31)	2.66 (1.16)	3.26 (1.01)	3.40 (0.77)	2.94 (0.89)
ApoB (mg/dL)	119.78 (18.27)	115.11 (17.41)	112.33 (22.06)	117.89 (13.77)	121.44 (15.82)	122.44 (15.63)
ApoA (mg/dL)	123.04 (16.95)	124.74 (18.61)	130.52 (15.26)	120.11 (22.63)	115.78 (22.94)	118.59 (18.92)
ApoB/ApoA	0.99 (0.21)	0.94 (0.18)	0.87 (0.21)	1.01 (0.21)	1.08 (0.24)	1.06 (0.20)
Systolic Blood Pressure (mmHg)	124.42 (6.68)	122.88 (7.77)	-	123.52 (7.31)	122.59 (5.61)	-
Diastolic Blood Pressure (mmHg)	81.54 (5.62)	80.38 (5.99))	-	80.37 (6.19)	79.44 (5.60)	-
AST (UI/L)	18.48 (5.14)	18.59 (3.96)	18.81 (3.29)	18.37 (3.94)	18.78 (3.88)	17.59 (3.46)
ALT (UI/L)	19.48 (5.06)	19.19 (4.35)	17.52 (3.75)	20.22 (3.84)	20.33 (3.23)	20.78 2.69
GGT (U/L)	18.19 (7.20)	18.30 (6.82)	19.30 (4.05)	19.44 (6.42)	19.22 (6.17)	19.00 (6.69)
Creatinine (mg/dL)	0.83 (0.07)	0.83 (0.08)	0.82 (0.07)	0.86 (0.12)	0.84 (0.12)	0.84 (0.12)
Waist Circumference (cm)	108.68 (10.79)	105.35 (9.81)	-	106.89 (12.63)	105.45 (12.19)	-
VAT (g)	1693.37 (871.50)	1482.63 (823.19)	-	1849.41 (851.68)	1861.04 (847.59)	-
Fat Mass (g)	32,620 (12,068.42)	30,847.37 (10,892.80)	-	36,380.78 (10,132.98)	34,940.78 (13,435.03)	-
Lean Mass (g)	43,296.41 (8917.54)	43,900.81 (9097.43)	-	45,429.04 (6867.50)	45,601.11 (7039.09)	-

- = no data were collected at this time point. Data are reported as mean (SD).

**Table 3 nutrients-12-03298-t003:** Between-group pre-post supplementation difference for primary and secondary endpoint. Estimate of time * treatment (β), 95% confidence interval (CI) and the adjusted *p*-value of the null hypothesis of a no effect are reported.

	Time * Group β (95% CI)	*p*-Value
***Primary endpoint***		
Fasting Glycemia (mmol/L)	0.36 (0.24; 0.48)	<0.0001
***Secondary endpoints***		
Total Cholesterol (mg/dL)	0.06 (−0.07; 0.14)	0.12
HDL (mg/dL)	−0.10 (−0.13; 0.06)	<0.0001
Total Cholesterol/HDL	0.27 (0.15; 0.40)	0.0002
LDL (mg/dL)	0.15 (0.05; 0.24)	0.005
LDL/HDL	0.23 (0.11; 0.35)	0.001
Triglycerides (mg/dL)	0.03 (−0.05; 0.12)	0.55
Insulin (mcU/mL)	1.28 (0.23; 2.33)	0.04
Glycated Hemoglobin (%)	0.21 (0.12; 0.31)	0.0002
A1c-Derived Average Glucose (mmol/L)	0.34 (0.18; 0.50)	0.0002
HOMA	0.55 (0.19; 0.91)	0.007
ApoB (mg/dL)	6.00 (1.69; 10.31)	0.01
ApoA (mg/dL)	−4.50 (−8.25; −0.75)	0.04
ApoB/ApoA	0.08 (0.03; 0.13)	0.003
Systolic Blood Pressure (mmHg)	0.61 (−2.47; 3.69)	0.83
Diastolic Blood Pressure (mmHg)	0.23 (−2.47; 2.92)	0.90
AST (UI/L)	−0.55 (−1.58; 0.47)	0.37
ALT (UI/L)	1.26 (0.24; 2.27)	0.03
GGT (U/L)	−0.78 (−1.94; 0.38)	0.28
Creatinine (mg/dL)	−0.002 (−0.02; 0.02)	0.90
Waist Circumference (cm)	1.89 (0.16; 3.61)	0.05
VAT (g)	222.37 (83.54; 361.20)	0.005
Fat Mass (g)	332.63 (−2754.20; 3419.46)	0.90
Lean Mass (g)	−432.33 (−1157.70; 293.03)	0.34

**Table 4 nutrients-12-03298-t004:** Within-group pre-post supplementation difference for primary and secondary endpoints. Estimate of time (β), 95% confidence interval (CI) and the adjusted p-value of the null hypothesis of a no effect are reported for the two groups.

	Supplemented Group		Placebo Group	
	Time β (95% CI)	*p*-Value	Time β (95% CI)	*p*-Value
***Primary endpoint***				
Fasting Glycemia (mmol/L)	−0.39 (0.31; 0.45)	<0.0001	−0.03 (−0.10; 0.04)	0.53
***Secondary endpoints***				
Total Cholesterol (mg/dL)	−0.03 (−0.09; 0.03)	0.003	0.03 (−0.004; 0.07)	0.38
HDL (mg/dL)	0.08 (0.06; 0.10)	<0.0001	−0.02 (−0.05; 0.11)	0.52
Total Cholesterol/HDL	−0.23 (−0.31; −0.15)	<0.0001	0.04 (−0.06; 0.14)	0.53
LDL (mg/dL)	−0.09 (−0.17; −0.01)	0.03	0.05 (−0.0004; 0.11)	0.30
LDL/HDL	−0.19 (−0.27; −0.11)	<0.0001	0.04 (−0.05; 0.13)	0.53
Triglycerides (mg/dL)	−0.04 (−0.12; 0.05)	0.41	−0.003 (−0.03; 0.03)	0.86
Insulin (mc/U/mL)	−1.76 (−2.73; −0.79)	0.001	−0.48 (−0.91; −0.05)	0.24
Glycated Hemoglobin (%)	−0.18 (−0.22; −0.13)	<0.0001	0.04 (−0.05; 0.13)	0.53
A1c-Derived Average Glucose (mmol/L)	−0.28 (−0.36; −0.20)	<0.0001	0.06 (−0.08; 0.20)	0.53
HOMA	−0.71 (−1.07; −0.41)	<0.0001	−0.16 (−0.37; 0.05)	0.48
ApoB (mg/dL)	−3.72 (−7.15; −0.30)	0.04	2.28 (−0.42; 4.97)	0.40
ApoA (mg/dL)	3.74 (2.19; 5.29)	<0.0001	−0.76 (−4.22; 2.70)	0.72
ApoB/ApoA	−0.06 (−0.09; −0.03)	0.002	0.02 (−0.01; 0.06)	0.52
Systolic Blood Pressure (mmHg)	−1.54 (−3.73; 0.66)	0.19	−0.92 (−3.19; 1.34)	0.53
Diastolic Blood Pressure (mmHg)	−1.15 (−2.70; 0.39)	0.18	−0.93 (−3.19; 1.34)	0.53
AST (UI/L)	0.17 (−0.65; 0.98)	0.68	−0.39 (−1.02; 0.25)	0.52
ALT (UI/L)	−0.98 (−1.79; −0.17)	0.03	0.28 (−0.35; −0.91)	0.53
GGT (U/L)	0.55 (−0.42; 1.54)	0.30	−0.22 (−0.87; 0.43)	0.57
Creatinine (mg/dL)	−0.006 (−0.02; 0.008)	0.42	−0.008 (−0.02; 0.006)	0.52
Waist Circumference (cm)	−3.33 (−4.67; −1.98)	<0.0001	−1.44 (−2.58; −0.29)	0.24
VAT (g)	−210.74 (−337.26; −84.22)	0.003	11.63 (−53.19; 76.55)	0.75
Fat Mass (g)	−1772.63 (−2857.07; −688.19)	0.003	−1440 (−4410.26; 1530.26)	0.53
Lean Mass (g)	604.41 (20.35; 1188.46)	0.05	172.07 (−287.25; 631.40)	0.54

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
