# Peer review of "The Metabolic Effects of Cynara Supplementation in Overweight and Obese Class I Subjects with Newly Detected Impaired Fasting Glycemia: A Double-Blind, Placebo-Controlled, Randomized Clinical Trial"

_nutrients, 2020, doi:10.3390/nu12113298_

Round 1

Reviewer 1 Report

This is an interesting study on cynara extract. 

Revision needed:

The phrase on row 45 is not clear. Please reformulate it.

In sample size section please add effect size to sample size calculation .

In tab 1 or in the results section please specify the gender distribution among the two groups.

Change definition of maximum and minimum blood pressure in Systolic and DIastolic

In the result section add the compliance to the treatment: were all compliant? how many tab did they take in %?

In the result section add the drop out % or specify that there were no dropout.

Did the subjects have adverse effects? If no, please specify that nobody have AE.

Author Response

To Editor-in-Chief,

We revised the manuscript with modifications and changes based on the reviewers’ comments.

We send you the revised manuscript together with our point-by-point response.

The changes in the text were highlighted in yellow.

Thanking you in advance for your kind collaboration and suggestions.

Best regards,

The authors

Reviewer 1

Keywords: It needs to be sorted in alphabetical order.

ANSWER: The keywords have been sorted in alphabetic order.

Authors should improve the introduction including the latest articles published for example in the MDPI platform or others, about other research in this field.

ANSWER: the introduction has been improved; new sentences and new references have been added

Methodology:
How many invitations did you send? Please write more about recruitment of the study group.
ANSWER: new sentences have been added

Did anyone drop out during the experiment?
ANSWER: Figure 1 presenting the patient flow chart has been added

How many tablets did the respondents receive? Did they all receive all tablets at once or in stages?

ANSWER: a new sentence has been added: “ the tablets (for the requirement of 4 weeks, 120 tablets in total) were delivered at the time of the first blood sample”

Did respondents return unused tablets?

ANSWER: At the end of the study, the patients returned the tablets. In the materials and methods, sentence has been revised in order to explain more clearly this topic.

Please provide the number consent of the Bioethics Committee

ANSWER: The number has been provided (9321/14122018).

There is no description of the strengths of the article. Authors should write about them in the end of discussion.

ANSWER: new sentences on strengths of the study have been added

Please check references as per the journal style.

ANSWER: The references have been adapted to journal style using a Reference Management Software.

Reviewer 2 Report

Thank you for the opportunity to review this article.

Comments and suggestions for Authors:

Keywords: It needs to be sorted in alphabetical order.
Authors should improve the introduction including the latest articles published for example in the MDPI platform or others, about other research in this field.

Methodology:
How many invitations did you send? Please write more about recruitment of the study group.
Did anyone drop out during the experiment?
How many tablets did the respondents receive? Did they all receive all tablets at once or in stages?
Did respondents return unused tablets?

Please provide the number consent of the Bioethics Committee

There is no description of the strengths of the article. Authors should write about them in the end of discussion.

Please check references as per the journal style.

Author Response

To Editor-in-Chief,

We revised the manuscript with modifications and changes based on the reviewers’ comments.

We send you the revised manuscript together with our point-by-point response.

The changes in the text were highlighted in yellow.

Thanking you in advance for your kind collaboration and suggestions.

Best regards,

The authors

Reviewer 2

This is an interesting study on cynara extract. 

Revision needed:

The phrase on row 45 is not clear. Please reformulate it.

ANSWER: the phrase has been reformulated

In sample size section please add effect size to sample size calculation .

ANSWER: We have added the effect size

In tab 1 or in the results section please specify the gender distribution among the two groups.

ANSWER: We have specified gender distribution among the two groups in the text

Change definition of maximum and minimum blood pressure in Systolic and Diastolic

ANSWER: the words have been changed

In the result section add the compliance to the treatment: were all compliant? how many tab did they take in %?

ANSWER: Figure 1 presenting the patient flow chart has been added. Moreover, a new sentence has been added on tablets.

In the result section add the drop out % or specify that there were no dropout.

ANSWER: there were no dropout. This sentence has been added

Did the subjects have adverse effects? If no, please specify that nobody have AE.

ANSWER: no subject had any adverse events. The sentence was added in the results and taken up in the discussion